# Multi-Tailed, Multi-Headed, Spatial Dynamic Memory Refined Text-to-Image Synthesis

## Abstract

Recent text-to-image generation methods that employ hundreds-of-millions to billions of model parameters or that are trained using tens-to-hundreds of GPUs of computational power have delivered highly-compelling and text-matching images. However, training to synthesize realistic images from text descriptions using a single-GPU resource remains a challenging task. In this paper we revisit the problem of generating images from text using models of less than 100 million parameters that can reliably be trained from scratch using a single-V100-GPU machine, and point out that there are still significant gains to be made within this problem setting. The current state-of-the-art amongst such low-resource models typically tackle text-to-image generation in a multistage manner, by first generating a rough initial image and then refining image details at subsequent stages. However, current methods suffer three important limitations. Firstly, initial images are generated at a sentence-level and provide a poor basis for word-level refinement. Secondly, by using common text-representations across all image regions, current refinement methods prevent different interpretations of words at different regions. Finally, images are refined in a single shot at each stage, limiting precision of image improvement. We introduce three novel components to address these shortcomings of low-resource methods: (1) A word-level initial stage to generate a better basis for refinement. (2) A spatial dynamic memory module to interpret words differently at different image regions. (3) An iterative multi-headed mechanism to better refine image details at each stage. We combine our three components as a unified model and demonstrate favourable performance against the previous state-of-the-art.

## 1 Introduction

Generative Adversarial Networks (GANs) have shown great promise for the generation of photo-realistic synthetic images (Goodfellow et al., 2014; Radford et al., 2015; Denton et al., 2015), and the success of GANs has driven research into conditional image-generation and multimodal learning. Text-to-image generation has especially received much of the community's attention and recent methods for text-to-image generation that employ very large models using hundreds-of-millions to billions of parameters or methods that require tens-to-hundreds of GPUs during training (Ramesh et al., 2021; Ding et al., 2021; Nichol et al., 2021; Rombach et al., 2022) have delivered highly-realistic text-matching images of general-scene content. However, while the results from these high-resource methods are highly compelling, their computational-requirements often make them prohibitively expensive to train-from-scratch in many settings. A key requirement for wider adoption of text-to-image generation technology on the other hand, remains that models be cheap to train-from-scratch for niche domains.

In this paper we revisit the problem of text-to-image generation using smaller low-resource models. In particular, we consider the setting where models employ less than 100 million parameters and can be trained from scratch using a single V100 GPU. We point out that there are still significant gains to be made by rethinking the design of such low-resource models - that are more practical to train than their high-resource counterparts.

We identify three important limitations in current state-of-the-art methods of lower-resource models (Xu et al., 2018; Li et al., 2019a; Zhu et al., 2019) and propose architectural solutions to address these problems. These lower-resource methods typically employ multiple-stages of image generation - conventionally, an initial image is first generated from a global sentence-level vector, and subsequent stages incorporate fine-grained information extracted from word-level vectors to refine image details. The first problem that we highlight is that by synthesizing image features only from a sentence-level vector, the initial generation stage often fails to provide a sufficient basis for word-level refinement. This is especially important for images possessing detailed spatial structure or co-dependent attributes. If co-dependent image details such as the head and a limb of an animal for example are poorly formed in the initial image, then their relative positions and details can be difficult to decide or correct during refinement at later stages. By using word-level information to synthesize images directly at the initial stage, we demonstrate that we are able generate more detailed initial images that provide a better basis for subsequent word-level refinement.

Secondly, current methods do not construct region-specific representations of text at refinement stages. This prevents us from interpreting the same words differently based on the content of image regions. Whereas, such requirement is natural in many contexts. The word 'vibrant' for example may dictate a requirement regarding the sharpness of a birds beak that is fundamentally different from the requirement that it dictates for the color of it's feathers. To better generate realistic images from natural text descriptions, it is important that we use a refinement architecture that allows different image regions to assimilate region-contextualized information from text.

Finally, we note that current methods generate refinement features (that modify previous image features) only once at each refinement stage and attempt to address all image aspects within a single-shot. This single-shot refinement limits the precision with which each refinement stage can learn to improve the prior image.

In this paper, we propose a Multi-Headed and Spatial Dynamic Memory image refinement mechanism with a Multi-Tailed Word-level Initial Generation stage (MSMT-GAN) to address these three issues. Our contributions are summarized as follows:

- We introduce a novel "Multi-Tailed" Word-level Initial Generation stage (MTWIG), that generates a separate set of image features for each word n-gram, and iteratively fuse these sets together to obtain better initial image features for subsequent refinement.

- We introduce a novel Spatial Dynamic Memory module (SDM) that fuses word-information in a custom way with each prior image region, to obtain region-contextualized representations of text. We extract refinement features from these SDM modules at each refinement stage.

- We introduce a novel Iterative Multi-Headed Mechanism (IMHM) of image refinement - wherein we explicitly allow each stage of refinement to make multiple distinct modifications to the prior image, under common discriminator feedback.

- We demonstrate that these three separate components work well together and that the addition of each component to the pipeline boosts generation performance on the Caltech-UCSD Birds 200 (CUB) dataset.

Experiment results demonstrate that in training-from-scratch, MSMT-GAN is competitive with current methods for low-resource models on the Microsoft Common Objects in Context (COCO) dataset (Lin et al., 2014) and significantly outperforms the previous state-of-the art for low-resource models on the CUB dataset, decreasing the lowest reported Fréchet Inception Distance (FID) (Heusel et al., 2017) by 21.58% and moving the R-precision (Xu et al., 2018) ahead by 8 standard deviations over the previous state-of-the-art for CUB.

We further note that while we have restricted our comparisons to models under 100 million parameters (that can reasonably be trained using a single V100 GPU), our method is only roughly half that size (Table 2).

## 2   Related Work

**Text-to-Image Generators:** Reed et. al. (Reed et al., 2016) first demonstrated that a translation model from natural language to image pixels could be learnt by conditioning both generator and discriminator networks of a GAN on input text-descriptions. There has since been a surge of interest in training multi-stage attention based GAN architectures for this task. While the conventional setting (Zhang et al., 2017; Xu et al., 2018; Li et al., 2019a; Zhu et al., 2019) assumes only the availability of (text,image) pairs at training time, recently a second setting has emerged that assumes availability of bounding-box/shape-mask information of objects attributes during training (Li et al., 2019b; Hinz et al., 2019; Cho et al., 2020; Liang et al., 2020). We highlight that this represents a significantly easier problem setting and that such methods are not feasible where bounding-box/shape information is unavailable (such as the CUB dataset). Our method does not assume the availability of bounding-box/shape information, and we make our comparisons against prior work of the same setting. As earlier mentioned, there has also been recent work that explores the use of very large models with billions of parameters for the task of text-to-image generation (Ramesh et al., 2021; Ding et al., 2021; Nichol et al., 2021). We do not make a comparison against these methods as our model tackles the problem of text-to-image generation using only a small fraction of their number of model parameters.

**Memory Networks:** Memory Networks (Weston et al., 2014) combine inference components with a long-term memory module that can be dynamically written to and read from. Current methods (Miller et al., 2016) query "key encodings" of memory slots to retrieve a set of weights. These weights are used to combine separate "value encodings" of the slots into a single response. A Dynamic Memory Generative Adversarial Network (DM-GAN) (Zhu et al., 2019) that retrieves information for image refinement from a memory module was proposed for text-to-image synthesis. In our SDM module, we too employ the *memory-writing, key-addressing, value-reading* paradigm introduced by (Miller et al., 2016), but our method differs from (Zhu et al., 2019) in all three memory operations. Fundamentally, DM-GAN does not create region-contextualized representations of text.

**Multi-Headed Attention:** Transformers (Vaswani et al., 2017) utilize a key-value mechanism similar to memory networks and introduced the idea of multi-headed attention. They linearly project query, keys and values to $h$ separate encodings, called "attention heads", and each head is separately used to extract an output vector. These vectors are concatenated together and linearly projected to a single response. Inspired by the success of Transformers, we introduce the IMHM method for image refinement. However, our method differs in a few respects. We maintain separate SDM modules for each head and we obtain queries and fuse outputs in an iterative fashion.

## 3   MSMT-GAN

Our MSMT-GAN architecture (Figure 1) comprises of three stages - a Multi-Tailed Word-level Initial Generation (MTWIG) stage, and two refinement stages. Each refinement stage is Multi-Headed, and each refinement head has a separate Spatial Dynamic Memory (SDM) module. The following sections present our MTWIG stage, our SDM module for a single refinement head, and the details of our Iterative Multi-Headed Mechanism (IMHM).

### 3.1   Multi-Tailed Word-level Initial Generation (MTWIG)

We highlight that previous multi-stage low-resource methods (Zhang et al., 2017; 2018; Li et al., 2019a; Zhu et al., 2019) all rely on the same type of initial generation stage and focus only on improving the refinement stages - making the conventional assumption that the performance of multi-stage generators is primarily determined by the refinement stages, and that the quality of the "rough initial image" is of little importance. In our paper, we break from this tradition and demonstrate for the first time that gains can be achieved in the final stage of image refinement by making an improvement to the initial images.

The conventional approach synthesizes initial images directly from a sentence-level vector without

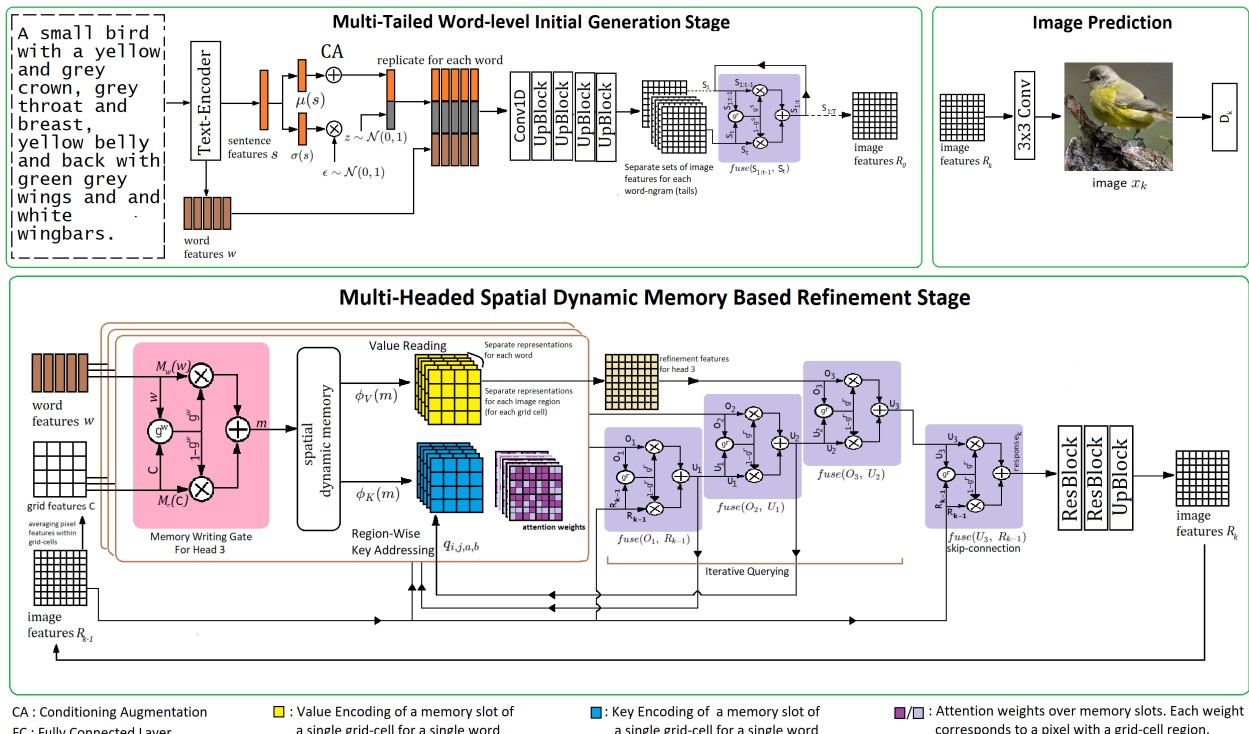

Figure 1: Our MSMT-GAN architecture for text-to-image synthesis, showing a Mutli-Tailed Word-level Initial Generation stage, a Multi-Headed Spatial Dynamic Memory based refinement stage with three refinement heads, and image prediction.

attempting to separate image attributes at a word-level. As a result, words-level details are are inherently excluded from the initial image. In our novel Multi-Tailed Word-level Initial Generation (MTWIG) stage, we overcome this shortcoming by explicitly creating separate sets of image attributes for each word n-gram.

First, we sample a vector of random noise $z$ from a normal distribution and use a pretrained text-encoder to extract a sentence-level vector and word-level vectors: $s$ and $W$ from the input text.

$$W = \{w_1, w_2, ..., w_L\}; \ w_l \ \in \ \mathbb{R}^{N_w} \quad s \ \in \ \mathbb{R}^{N_s} \quad ; \quad z_n \ \sim \ \mathcal{N}(0,1); z \ \in \ \mathbb{R}^{N_z} \quad (1)$$

Where $L$ is the number of words in the text-description, and $N_z$, $N_s$ and $N_w$ are the dimensions of the noise vector, sentence vector and word vectors respectively. To mitigate over-fitting, the Conditioning Augmentation technique (Zhang et al., 2017) is used to resample the sentence-vector from an independent Gaussian distribution. This resampled sentence vector $s'$ and the noise vector $z$ are concatenated with each word-level vector $w_l$ from the input text sequence, and the sequence of concatenated vectors are passed through a 1D convolutional operation $V$ of stride 1 (see Figure 1).

$$F = V(\{concat(s', \ z, \ w_l) \mid \ \forall \ w_l \in W\}) \quad (2)$$

The length $T$ of the output sequence $F$ depends on the kernel size used by $V$ and the vectors of the output sequence $f_t \in F$ are each separately passed through a series of upsampling blocks to generate corresponding sets of image features $S_t$. These sets of image features or "tails" each correspond to a different word n-gram from the input text sequence. If we use a kernel size of 1 for $V$, then each tail $S_t$ corresponds to a single word. If we use a kernel size of 2, then each tail $S_t$ corresponds to a word bi-gram, and so on. We combine our sequence of tails $\{S_t\}$ together in an iterative fashion using the adaptive gating fusion mechanism introduced by (Zhu et al., 2019) (discussed in Appendix).

$$S_{1:t} = fuse(S_{1:t-1}, \ S_t, \ P^{\text{MTWIG}}, \rho^{\text{MTWIG}}) \ ; \ R_1 = S_{1:T} \quad (3)$$

Where $P^{\mathrm{MTWIG}}$ and $\rho^{\mathrm{MTWIG}}$ denote parameter matrix and bias terms, $S_{1:t}$ denotes a combination of the first $t$ tails, and $S_{1:1}$ denotes the first tail $S_1$. The combination of all $T$ tails gives us the final image features $R_1$ for our initial stage. Notice that by concatenating each word vector $w_l$ with $s'$ and $z$ before the 1D convolution, each tail is created with some common information, so they may learn to fuse together coherently. Each upsampling block consists of a nearest neighbor upsampling layer and a 3×3 convolution operation. An initial image is predicted from $R_1$ using a 3×3 convolution.

## 3.2 Spatial Dynamic Memory (SDM)

In this section, we describe the operation of a single refinement head. Unlike previous methods, our novel Spatial Dynamic Memory (SDM) module creates a separate region-contextualized text representations for each image region. This allows us to interpret the same text in fundamentally different ways at different parts of an image and assimilate region-contextualized information from text at each part. To begin with, we have the set of word-level vectors $W$ and image features $R_{k-1}$ from the previous stage of generation.

$$R_{k-1} = \{r_{1,1}, \ r_{1,2}, \ ..., \ r_{s,s}\}; \ r_{u,v} \in \mathbb{R}^{N_r} \tag{4}$$

Where $|s \times s|$ is the number of image pixels and $N_r$ is the dimension of pixel features. We obtain refinement features in three steps: *Memory Writing, Key Addressing* and *Value Reading.*

**Memory Writing**: First, we divide the fine-grained $s \times s$ inital image into a coarse $h \times h$ sized grid-map and average the pixel features within each grid-cell to get grid-level image features $C$.

$$C_{i,j} = \frac{1}{|p \times p|} \sum_{u=(i-1)*p+1}^{i*p} \sum_{v=(j-1)*p+1}^{j*p} r_{u,v} \tag{5}$$

Where $p = s/h$, so that $|p \times p|$ are the number of pixels represented by each grid cell. Then, we create $L \times h \times h$ memory slots $\{m_{l,i,j}\}$ - one corresponding to each word $l$ for each grid-cell $(i,j)$. These slots are our region-contextualized representations of each word, and each slot uses a separate memory writing gate $g^w_{l,i,j}$ to fuse information from each grid-cell $(i,j)$ with each word feature $w_l$.

$$g^w_{l,i,j}(R_{k-1}, w_L) = \sigma\left(A * w_l + B_{i,j} * C_{i,j}\right) \tag{6}$$

$$m_{l,i,j} = M_w(w_l) \odot g^w_{l,i,j} + M_c(C)_{i,j} \odot (1 - g^w_{l,i,j}) \tag{7}$$

The grid-level features $C$ are encoded using a 2d convolution operation $M_c$ (with stride 1 and $N_m$ output filters) and we use a common 1x1 convolution operation $M_w$ to encode all word vectors to a $N_m$ dimensional space. $A$ and $B_{i,j}$ are $1 \times N_w$ and $1 \times N_r$ matrices respectively.

**Key Addressing:** In this step, we compute attention weights $\{\alpha_{l,i,j,a,b}\}$ over our region-contextualized text-representations $\{m_{l,i,j}\}$. The dimensions $(a,b)$ index pixels within grid-cell $(i,j)$, so that each slot $m_{l,i,j}$ gets a matrix $\alpha_{l,i,j} : p \times p$ of attention weights. Each weight is computed as a similarity probability between a key-encoding of the slot: $\phi_K(m_l)_{ij}$ and a query vector: $q_{i,j,a,b}$, where $\phi_K(.)$ is a 2d conv. operation of stride 1 and $(N_r + p^2)$ output filters.

$$\alpha_{l,i,j,a,b} = \frac{\exp(\phi_K(m_l)_{i,j} * q_{i,j,a,b})}{\sum_{l=1}^{L} \exp(\phi_K(m_l)_{i,j} * q_{i,j,a,b})} \tag{8}$$

In the case of single headed image refinement, we use the previous image features $R_{k-1}$ to obtain the query vectors. A query vector $q_{i,j,a,b}$ is made up of three components, 1)A global-level query: $q^{global}$, 2)A grid-level query: $q^{grid}_{i,j}$, and 3)A pixel-level query: $q^{pixel}_{i,j,a,b}$. To obtain these three components, we encode $R_{k-1}$ using three separate 2d convolution operations: $\phi_{Q^{global}}(.)$, $\phi_{Q^{grid}}(.)$ and $\phi_{Q^{pixel}}(.)$, each with a stride of 1 and $N_r$ output filters.

$$Q^{global} = \phi_{Q^{global}}(R_{k-1}) \quad ; \quad Q^{grid} = \phi_{Q^{grid}}(R_{k-1}) \quad ; Q^{pixel} = \phi_{Q^{pixel}}(R_{k-1}) \tag{9}$$

Then, the average of all pixel features of $Q^{global}$ becomes the global-level query component $q^{global}$. The average of pixel features within the grid cell $(i,j)$ of $Q^{grid}$ becomes the grid-level query $q^{grid}_{i,j}$, and the

pixel feature at location $(a, b)$ within grid cell $(i, j)$ is extracted from $Q^{pixel}$ to give us the pixel-level query component $q^{pixel}_{i,j,a,b}$.

$$q^{global} = \frac{1}{|s \times s|} \sum_{u=1}^{s} \sum_{v=1}^{s} Q^{global}_{u,v} \quad ; \tag{10}$$

$$q^{grid}_{i,j} = \frac{1}{|p \times p|} \sum_{u=(i-1)*p+1}^{i*p} \sum_{v=(j-1)*p+1}^{j*p} Q^{grid}_{u,v} \tag{11}$$

$$q^{pixel}_{i,j,a,b} = Q^{local}_{h(i,a),\ h(j,b)} \quad ; \quad h(i,a) = (i-1)*p+a \tag{12}$$

Where $(h(i,a),\ h(j,b))$ indexes the pixel at location $(a, b)$ within grid-cell $(i, j)$. To obtain the final query $q_{i,j,a,b}$, we concatene these three components together.

**Value Reading:** In the value reading step, for each pixel $(a, b)$ within a grid-cell $(i, j)$, we compute a weighted sum of value-encoded memory slots: $\phi_V(m_l)_{ij}$ along the word dimension $l$.

$$e_{i,j,a,b} = \sum_{l=1}^{L} \alpha_{l,i,j,a,b} \cdot \phi_V(m_l)_{i,j} \tag{13}$$

$\phi_V(.)$ is a 2d convolution operation with stride 1 and $N_r$ output filters. We now have $e_{i,j} : p \times p \times N_r$ dimensional matrices - each of which corresponds to a single grid cell of our coarse $h \times h$ grid map. To obtain $s \times s$ fine-grained refinement features, we apply the mapping:

$$o_{h(i,a),\ h(j,b)} = e_{i,j,a,b} \tag{14}$$

Where $h(.,\ .)$ is the function defined in Eq.12. That is, we populate each grid cell with $|p \times p|$ vectors of $N_r$ dimensionality. Since $p = s/h$, we are left with a set of refinement features $O = \{o_{u,v}\}$ that are made up of $|s \times s|$ vectors of $N_r$ dimensionality, each corresponding to a single pixel.

### 3.3 Iterative Multi-Headed Mechanism (IMHM)

Current methods generate refinement features only once at each refinement stage and attempt to address all image aspects in a single-shot. This limits the precision with which each refinement stage can learn to improve the prior image. In order to make it easier for each refinement stage to precisely address multiple image aspects, we introduce a novel iterative multi-headed mechanism that makes multiple distinct modifications to the prior image features under common discriminator feedback. Each head of our mechanism has a separate spatial dynamic memory module formed from $R_{k-1}$ and $W$. For the first refinement head, we use the previous image features $R_{k-1}$ to obtain a query matrix and extract a set of refinement features $O_1$ exactly as described in Section 3.2. Then, we fuse $O_1$ and $R_{k-1}$ using the fusion mechanism introduced by (Zhu et al., 2019) (described in Section 7.1) to obtain an updated set of image features $U_1$. If we use only a single refinement head, then this becomes our response for the refinement stage $k$. However, if we use more than one refinement head, then for the next head, we use $U_1$ to obtain a query matrix. That is, we follow the same mechanism outlined in Section 3.2, but replace $R_{k-1}$ with $U_1$ in Eq.9. Doing so, we extract a second set of refinement features $O_2$, and we fuse $O_2$ and $U_1$ to obtain updated image features $U_2$. We proceed in this iterative fashion until we have used all of our refinement heads. The final updated image features are fused with the original image features $R_{k-1}$ in a skip-connection to obtain the final response of the refinement stage $k$. That is, if we have $T$ refinement heads:

$$U_t = fuse(U_{t-1},\ O_t,\ P_t,\ \rho_t) \tag{15}$$

$$response_k = fuse(U_T,\ R_{k-1},\ P^{skip},\ \rho^{skip}) \tag{16}$$

Notice, we use separate parameter matrix and and bias terms $P$ and $\rho$ for each fusion operation, so that we combine refinement features and image features in a custom way for each head. The, $response_k$ is passed through several residual blocks (He et al., 2016) and an upsampling block to obtain higher resolution image features $R_k$. Each block consists of a nearest neighbor upsampling layer and a $3 \times 3$ convolution operation. Finally, a refined image $x_k$ is predicted from $R_k$ using a $3 \times 3$ convolution operation.

# 4 Experiments

**Datasets:** We evaluate our method on the Caltech-UCSD Birds 200 (CUB) dataset (Wah et al., 2011) and the Microsoft Common Objects in Context (COCO) dataset (Lin et al., 2014). The CUB dataset contains 8,855 training images and 2,933 test images, with 10 corresponding text descriptions for each image. The COCO dataset, contains 82,783 training images and 40,504 test images, with 5 corresponding text descriptions for each image. We preprocess the datasets according to the methods introduced by (Zhang et al., 2017).

**Evaluation metrics:** To evaluate the realism of images, we rely on the Fréchet Inception Distance (FID) (Heusel et al., 2017). FID computes the distance between synthetic and real-world image distributions based on features extracted from a pre-trained Inception v3 network. A lower FID indicates greater image-realism. To evaluate the relevance of a synthetic image to it's generating text-description, we rely on the R-precision introduced by (Xu et al., 2018). R-precision is computed as the mean accuracy of using each synthetic image to retrieve one ground truth text-description from among 100 candidates. To evaluate a model on a dataset, we generate 30,000 synthetic images conditioned on text descriptions from the unseen test set.

**Implementation Details:** To obtain a sentence-level vector and word-level vectors for a given text description, we use the pretrained bidirectional LSTM text encoder employed by AttnGAN (Xu et al., 2018). Our MTWIG stage synthesizes images features with 64x64 resolution. Two refinement stages refine these features to 128x128 and 256x256 resolution respectively. At refinement stages, we use $T = 6$ refinement heads and we use $h = 8$ to divide each input image into a coarse $8 \times 8$ grid map. We use the same discriminator network architecture employed by (Zhu et al., 2019). Our objective Function is largely the same as that used by previous methods (Xu et al., 2018; Li et al., 2019a; Zhu et al., 2019) with the addition of a redundancy loss to encourage each head of refinement to focus on different image aspects. Further implementation details are provided in the Appendix.

## 4.1 Ablative Experiments

**Effectiveness of Multi-Tailed Word-level Initial Generation:** In our experiments (Appendix), we find that our MTWIG stage is most effective for single stage generation when used with a kernel size of 3, so that we generate a separate tail for each word tri-gram. To evaluate the effectiveness of our MTWIG(ks=3) stage in the context of multi-stage models, we train our MTWIG(ks=3) method with DM-GAN (Zhu et al., 2019) style-refinement stages for 700 epochs on the CUB dataset, and observe that it provides a better basis for word-level refinement, decreasing the FID score achieved by the original DM-GAN model by 7.72% and increasing R-precision by 2.76% (Table 1). Figure 2 shows the improved visual quality of the refined images. We again point out that previous multi-stage methods (Zhang et al., 2017; 2018; Xu et al., 2018; Li et al., 2019a; Zhu et al., 2019) all rely on the same type of initial generation stage, and we expect a similar boost in performance if we replace their initial stage with ours.

**Effectiveness of Spatial Dynamic Memory:** In order to evaluate the effectiveness of our SDM based-refinement stages, we compare a multi-stage model that uses our MTWIG(ks=3) stage and DM-GAN's refinement stages against a multi-stage model that uses our MTWIG(ks=3) stage and our single-headed SDM-based refinement stages. Both models are trained on the CUB dataset for 700 epochs. We observe that our SDM-based refinement out-performs DM-GAN's refinement, decreasing FID score by 4.64% , and boosting R-precision by an additional 0.83% (Table 1). Figure 2 shows that SDM-based refinement generates images of better visual quality than DM-GAN's refinement stages for the same initial generation architecture.

**Effectiveness of the Iterative Multi-Headed Mechanism:** To evaluate the effectiveness of our IMHM refinement, we compare the model that uses MTWIG(ks=3) and single-headed SDM-based refinement stages against our full MSMT-GAN model - that uses MTWIG(ks=3) and six SDM-based refinement heads for each stage. As before, both models are trained for 700 epochs on the CUB dataset. We find that refinement stages that use our multi-headed mechanism out-perform single-headed refinement stages, decreasing FID score 2.38%, and boosting R-precision by another 1.03% (Table 1). Visually, we find that

images generated by our IMHM refinement posses text-irrelevant content that is far more detailed than that observed in images generated by single-headed refinement stages (Figure 2).

Table 1: FID and R-precision of DM-GAN and ablative versions of our model. (With all of our variants trained for 700 epochs on CUB)

| Method | CUB | |
| --- | --- | --- |
| | FID ↓ | R-prcn (%) ↑ |
| DM-GAN | 11.91 | 76.58 ± 0.53 |
| MTWIG w/ DM-GAN's refinement | 10.99 | 79.37 ± 0.73 |
| MTWIG w/ SDM refinement | 10.48 | 80.20 ± 0.67 |
| MTWIG w/ SDM and IMHM refinement (MSMT-GAN) | **10.23** | **81.23 ± 0.68** |

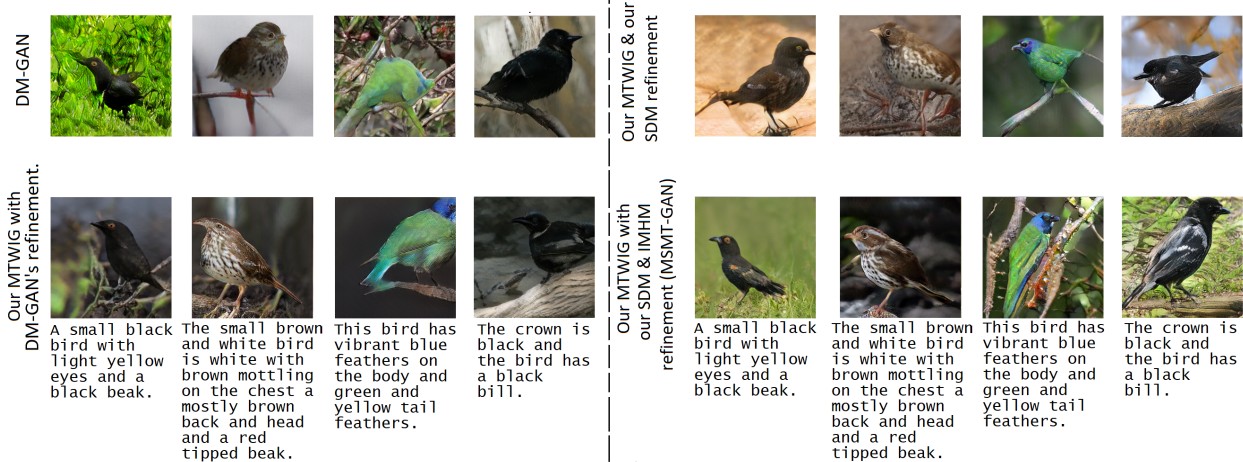

Figure 2: Comparison of DM-GAN with ablative versions of our model trained for 700 epochs on the CUB dataset.

Table 2: Number of parameters required at test-time (including text-encoders) for the previous state-of-the-art in comparison to our MSMT-GAN (approximate values reported in millions).

| Method | #Param. CUB | #Param. COCO | #Train V100 GPUs required |
| --- | --- | --- | --- |
| AttnGAN | 9.16M | 22.43M | 1 |
| ControlGAN | 30.72M | 45.36M | 1 |
| DM-GAN | 23.52M | 30.96M | 1 |
| DF-GAN | 14.32M | 20.87M | 1 |
| Our MSMT-GAN | 48.11M | 55.16M | 1 |
| XMC-GAN | - | >111M | >3 |

## 4.2 Comparison with State of the Art

To compare our architecture against the previous state-of-the-art for the task of text-to-image generation, we train MSMT-GAN models for 1000 epochs on the CUB dataset and for 210 epochs on the COCO dataset. As shown in Table 3, our MSMT-GAN decreases the previous lowest reported FID score by 21.58% on the CUB dataset -marking a significant improvement in image realism, and also boosts the previous best

reported CUB R-precision by 4.24% (placing it 8 standard deviations ahead of the previous state-of-the-art) - marking a large improvement in the similarity of synthetic images to their generating text. As shown in Table 2 and Table 3, our model is comparable in size to previous methods, and outperforms the next closest contender of similar size for COCO (DM-GAN) by 4.21% on FID score -making it highly competitive with the current state-of-the-art for COCO. We also observe a slight improvement of 0.23% on COCO R-precision. Qualitatively, we observe that synthetic images generated by our model are typically sharper and more realistic than those generated by prior methods (Figure 3). In particular, we observe that our method generates synthetic images that possess greater detail and that are better matches for the generating text.

Table 3: FID and R-precision of the previous state-of-the-art and our MSMT-GAN trained for 1000 epochs on CUB and 210 epochs on COCO.

| | CUB | | COCO | |
| --- | --- | --- | --- | --- |
| Method | FID ↓ | R-prcn (%) ↑ | FID ↓ | R-prcn (%) ↑ |
| AttnGAN[1] | 14.01 | 67.82 ± 4.43 | 29.53 | 85.47 ± 3.69 |
| ControlGAN | - | 69.33 ± 3.23 | - | 82.43 ± 2.43 |
| DF-GAN[1] | 13.48 | - | 33.29 | - |
| DM-GAN[1] | 11.91 | 76.58 ± 0.53 | 24.24 | 92.23 ± 0.37 |
| Our MSMT-GAN | **9.34** | **80.82 ± 0.54** | **23.22** | **92.46 ± 0.28** |

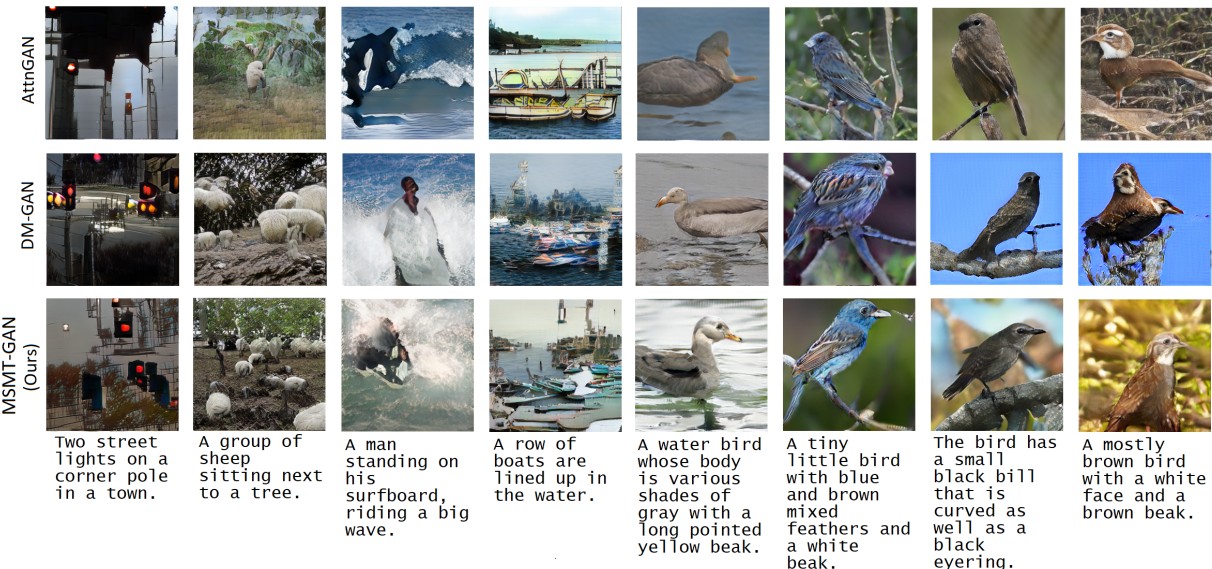

Figure 3: Comparison of MSMT-GAN with state-of-the art models on the CUB and COCO datasets.

We attribute the larger improvement for CUB over COCO to the greater spatial-structure of image attributes in CUB images over the loosely coupled nature of objects in the COCO dataset, and point out that that accounting for detailed spatial structure over image attributes is a key issue tackled by our MTWIG and SDM modules.

---

[1]We make our comparisons against pretrained models released by the authors, and we report results using the official implementations of FID score. Note on evaluation metrics in Appendix.

## 5    Broader Impact

Similar to other GAN models, we reflect that one may synthesize realistic-looking artificial data to maliciously fool human-beings, and lower-resource models makes it potentially easier for offenders to do so. However, we also note that the ability to quickly and cheaply train text-to-image models from scratch with better quality for niche domains offers significant benefits to human-computer-interaction and computer-aided-design, democratizing a technology that otherwise might be limited to those few groups with high-resources.

## 6    Conclusion

In this work, we revisited the problem of training text-to-image generation models from scratch using a single-V100-GPU and less than 100 million network parameters, pointing out that there are still significant gains to be made by rethinking the design of such methods. We identified three limitations of previous low-resource models and proposed the unified MSMT-GAN architecture to address them. First, we introduced a novel Multi-Tailed Word Level Initial Generation stage (MTWIG) that explicitly generates separate sets of image features for each word n-gram. Second, we proposed a novel Spatial Dynamic Memory (SDM) module to contextualize text representations by image-region. Third, we introduced a novel Iterative Multi-Headed Mechanism (IMHM) of image refinement to make it easier for each refinement stage to precisely address multiple image aspects. Our ablative experiments demonstrate that these three separate components work well together and that the addition of each component to the pipeline boosts generation performance on the CUB dataset. Benchmarking experiments further demonstrate that our MSMT-GAN model significantly out-performs the previous state of the art amongst low-resource models on the CUB dataset, decreasing the lowest reported FID score by 21.58% and boosting R-precision ahead by 8 standard deviations over the previous state-of-the-art. On the COCO dataset, we have demonstrated that MSMT-GAN is highly competitive with current methods based on image realism and model resources. In future work, we aim to design a discriminator model that provides more region-specific feedback than existing methods, to use in conjunction with our MSMT-GAN.

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

# 7 Appendix

## 7.1 Fusing two sets of image features together

Given any two sets of image/refinement features $A = \{a_{u,v}\}$ and $B = \{b_{u,v}\}$ such that $a_{u,v}, b_{u,v} \in \mathbb{R}^{N_r}$, we can use the adaptive gating mechanism introduced by (Zhu et al., 2019) to obtain a combined/updated set of features $\{r_{u,v}\}$, where $r_{u,v} \in \mathbb{R}^{N_r}$.

$$g_{u,v}^r = \sigma(P * concat(a_{u,v},\ b_{u,v}) + \rho) \tag{17}$$

$$r_{u,v} = a_{u,v} \odot g_{u,v}^r + b_{u,v} \odot (1 - g_{u,v}^r) \tag{18}$$

Where $(u, v)$ index pixel locations, $N_r$ is the dimension of image pixel features, $g_{u,v}^r$ is a gate for information fusion, $\sigma$ is the sigmoid function and $P$ and $\rho$ are the parameter matrix and bias term respectively. In our paper, we invoke this adaptive gating mechanism by a function "$fuse$" such that $fuse(A, B, P, \rho) = \{r_{u,v}\}$.

## 7.2 Analysis of Multi-Tailed Word-level Initial Generation

Table 4: Quantitative comparison of Fréchet Inception Distance and R-precision between previous Initial Generation (IG) and our Multi-Tailed Word-level Initial Generation (MTWIG) stage for varied kernel sizes (ks) on the CUB and COCO datasets.

| Method | CUB | | COCO | |
|---|---|---|---|---|
| | FID ↓ | R-prcn (%) ↑ | FID ↓ | R-prcn (%) ↑ |
| IG | 119.39 | **83.54 ± 0.57** | 51.22 | 94.57 ± 0.53 |
| MTWIG (ks=1) | 118.49 | 82.78 ± 0.58 | 41.79 | 94.47 ± 0.49 |
| MTWIG (ks=2) | 120.76 | 82.82 ± 0.92 | 40.23 | 94.65 ± 0.39 |
| MTWIG (ks=3) | **115.38** | 82.94 ± 0.74 | **39.85** | **94.80 ± 0.29** |

To analyze the effectiveness of our MTWIG stage, we make comparisons with the inital generation (IG) stage employed by previous methods (Zhang et al., 2017), (Zhang et al., 2018), (Xu et al., 2018), (Li et al., 2019a), (Zhu et al., 2019). We train IG and MTWIG stages without refinement for 300 epochs on the CUB dataset and for 60 epochs on the COCO dataset. Table 4 shows a quantitative comparison between IG and MTWIG architectures that use different kernel sizes. We observe that our MTWIG method achieves best results for ks=3 (that is, by forming separate image-feature sets for word tri-grams), decreasing FID scores from the previous IG method by 3.36% on the CUB dataset and by 22.2% on the COCO dataset. In single-stage generation, the larger improvement observed on the COCO dataset highlights that our MTWIG method is most beneficial for complex scene generation, where the presence of a large number of distinct objects

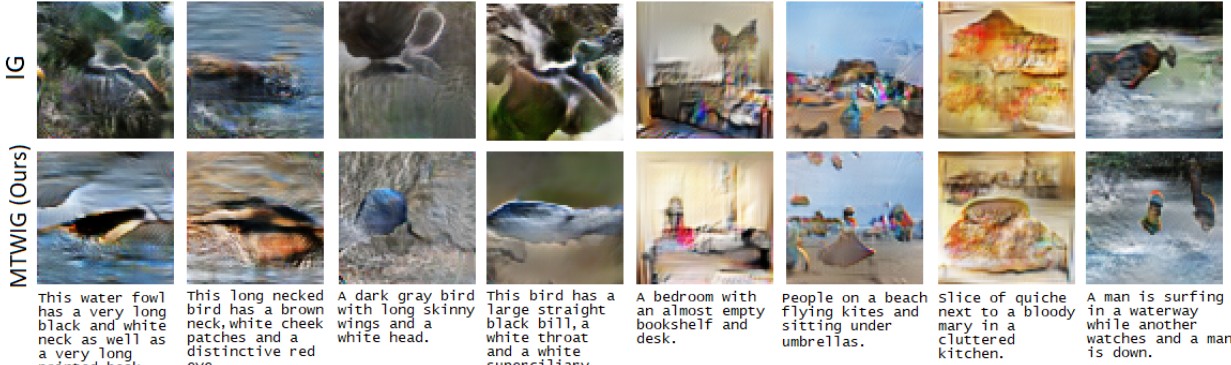

Figure 4: Multi-Tailed Word-level Initial Generation (MTWIG; kernel size=3) in comparison to conventional Initial Generation (IG).

demands word-level separation of attributes. Our MTWIG stages also achieves competitive R-precision scores on both datasets, demonstrating that images synthesized by our method are well conditioned on their input text-descriptions. In Figure 4, we visually compare 64x64 images generated by IG and images generated by our MTWIG(ks=3) model. We observe that images generated by our method typically posses object attributes that are better separated and more clearly discernible.

### 7.3 Additional Implementation Details

We set $N_w = 256$, $N_r = 48$ and $N_m = 96$ to be the dimension of text, image and memory feature vectors respectively. We set the hyperparameters $\{\lambda_1 = 1, \lambda_2 = 0.5, \lambda_3 = 5\}$ for the CUB dataset and $\{\lambda_1 = 1, \lambda_2 = 0.5, \lambda_3 = 50\}$ for the COCO dataset. All networks are trained using an ADAM optimizer (Kingma & Ba, 2015) with $\beta_1 = 0.5$ and $\beta_2 = 0.999$. The learning rate is set to be 0.0002 and we use a batch size of 20 for all networks. We train our models on a single Tesla V100 gpu, and observe that similar to previous multi-stage methods, training our MSMT-GAN method is a GPU intensive process. We require approximately 27min for one epoch of the CUB dataset, and approximately 4.7 hours for one epoch of the COCO dataset.

### 7.4 Objective Function

The objective function for our generator network is defined as:

$$L = \lambda_1 L_{CA} + \sum_{k=2}^{K} \lambda_2 L_{RED_k} + \sum_{k=1}^{K} (L_{G_k} + \lambda_3 L_{DAMSM_k}) \tag{19}$$

Where, $L_{CA}$ denotes the conditioning augmentation loss (Zhang et al., 2017), $G_k$ denotes the generator of the $k^{th}$ stage so that $L_{G_k}$ denotes the adversarial loss for $G_k$, $L_{RED_k}$ denotes our redundancy loss for the $k^{th}$ stage, and $L_{DAMSM_k}$ denotes the DAMSM text-image matching loss (Xu et al., 2018) for the $k^{th}$ stage. $\lambda_1$, $\lambda_2$ and $\lambda_3$ are hyperparameters that combine the various losses.

**Redundancy Loss:** To encourage each head of a refinement stage to focus on separate image aspects, we average region-wise information of each head's output refinement features and penalize similarity between different refinement heads. That is, for $T$ refinement heads:

$$f(t) = \frac{1}{|s \times s|} \sum_{u=1}^{s} \sum_{v=1}^{s} o_{u,v} \tag{20}$$

$$L_{RED_k} = \sum_{i=1}^{T} \sum_{j=i+1}^{T} sim(f(i), f(j)) \tag{21}$$

Where $o_{u,v} \in O_t$ in stage $k$ (see Section 3.3) and $sim$ is the cosine similarity between vectors. We call this sum of pairwise similarity $L_{RED_k}$ the "redundancy loss" of the $k^{th}$ refinement stage.

**Adversarial Loss:** The adversarial loss for $G_k$ is defined as:

$$L_{G_k} = -\frac{1}{2}[\mathbb{E}_{x \sim p_{G_k}} \log D_k(x) + \mathbb{E}_{x \sim p_{G_k}} \log D_k(x, s)] \tag{22}$$

$D_k$ is the discriminator network for the $k^{th}$ stage of generation. The first term provides feedback on image realism independent of the input text, and the second term provides feedback on the realism of the image in light of the input text. Alternate to adversarial training of $G_k$, each discriminator $D_k$ is trained to classify images as real or fake by minimizing the discriminator loss $L_{D_k}$.

$$L_{D_k} = \underbrace{-\frac{1}{2}[\mathbb{E}_{x \sim p_{data}} \log D_k(x) + \mathbb{E}_{x \sim p_{G_k}} \log(1 - D_k(x))]}_{\text{unconditional loss}} + \underbrace{-\frac{1}{2}[\mathbb{E}_{x \sim p_{data}} \log D_k(x, s) + \mathbb{E}_{x \sim p_{G_k}} \log(1 - D_k(x, s))]}_{\text{conditional loss}} \tag{23}$$

Where the unconditional component distinguishes synthetic and real images independent of the input text, and the conditional component distinguishes them in light of the input text.

### 7.5 Note on Evaluation Metrics

We point out that there is inconsistency in the FID implementation used to evaluate prior methods. While some methods (Li et al., 2019b) report scores using the official Tensorflow version of FID - which uses the weights of a pretrained Tensorflow inception model, other methods (Zhu et al., 2019; Zhang et al., 2021) have reported scores using an unofficial Pytorch implementation of FID - that uses the weights of a pretrained Pytorch inception model.

Each of these implementations computes different values of FID scores for the same sets of images, and scores computed from the two versions are **not** directly comparable with each other. We highlight that the correlation between FID score and human judgement is an empirical observation, that is dependent on the weights of the pretrained inception model used. The correlation with human judgement has **only** been shown to hold true for the weights of the pretrained Tensorflow inception model (used by the FID authors -(Heusel et al., 2017)), and has **not** been verified for the unofficial Pytorch inception model (which has different weights). As such, to ensure that we correlate with human judgement, in our paper we only report scores using the official Tensorflow implementation of FID score - computing values for prior work from the pretrained models released by their authors. In Table 5 below, we additionally report Pytorch-implementation FID scores of SEGAN (Tan et al., 2019) and XMC-GAN (Zhang et al., 2021) models, as reported by their authors. It was not possible for us to recompute these scores using the official Tensorflow version of FID as pretrained models for these methods have not been made publicly available.

We again highlight, that the FID scores computed by the official Tensorflow implementation are not directly comparable with the scores computed by the unofficial Pytorch implementation.

In Table 5, we additionally benchmark the performance of models on the Inception Score (IS) (Goodfellow et al., 2014). We note however, that the FID score (which is reference-based and compares the distributions of real and synthetic images together) has been observed to be more consistent with human judgement of image realism than IS (which is reference-free and does not make comparisons to real images) (Heusel et al., 2017). We were not able to recompute other metrics for RiFeGAN (Cheng et al., 2020) and LeciaGAN (Qiao et al., 2019) as the pretrained models have not been made publicly available.

In Table 5, "-" represents cases where the data was not reported or is reported in a manner which is non-comparable (besides FID values) and we note that methods which appear in the comparison for one dataset (CUB/COCO) and not for another reported no data for the missing dataset or reported no data that can be readily compared.

---

[1]We make our comparisons against the pretrained models released by the authors, and we report results using the official implementations of FID score.

[3] We use the FID score reported by the paper, but note that this was computed using an unofficial Pytorch implementation of FID - which is not directly comparable with the official FID implementation scores. See Section 7.5 for further details.

Table 5: FID and R-precision and IS of the previous methods and our MSMT-GAN trained for 1000 epochs on CUB and 210 epochs on COCO.

| Method | CUB | | | COCO | | |
|---|---|---|---|---|---|---|
| | FID ↓ | R-prcn (%) ↑ | IS ↑ | FID ↓ | R-prcn (%) ↑ | IS ↑ |
| AttnGAN[1] | 14.01 | 67.82 ± 4.43 | 4.36 ± 0.03 | 29.53 | 85.47 ± 3.69 | 25.89 ± 0.47 |
| ControlGAN | - | 69.33 ± 3.23 | 4.58 ± 0.09 | - | 82.43 ± 2.43 | 24.06 ± 0.60 |
| SEGAN | 18.16[3] | - | 4.67 ± 0.04 | 32.28 | - | 27.86 ± 0.31 |
| RiFeGAN | - | - | **5.23 ± 0.09** | - | - | - |
| LeciaGAN | - | - | 4.62 ± 0.06 | - | - | - |
| DM-GAN[1] | 11.91 | 76.58 ± 0.53 | 4.71 ± 0.06 | 24.24 | 92.23 ± 0.37 | **32.43 ± 0.58** |
| DF-GAN[1] | 13.48 | - | 4.70 ± 0.06 | 33.29 | - | 18.64 ± 0.64 |
| XMC-GAN | - | - | - | **9.33**[3] | - | 30.45 |
| Our MSMT-GAN | **9.34** | **80.82 ± 0.54** | 4.55 ± 0.06 | 23.22 | **92.46 ± 0.28** | 28.91 ± 0.35 |

## 7.6 More example images

Figures 5 and 6 below show more images generated by our method in comparison to prior work on the CUB and COCO datasets. Again, we observe that synthetic images generated by MSMT-GAN are typically sharper and more realistic than those generated by previous methods. We also observe that the synthetic images generated by MSMT-GAN are usually a better match for the generating text.

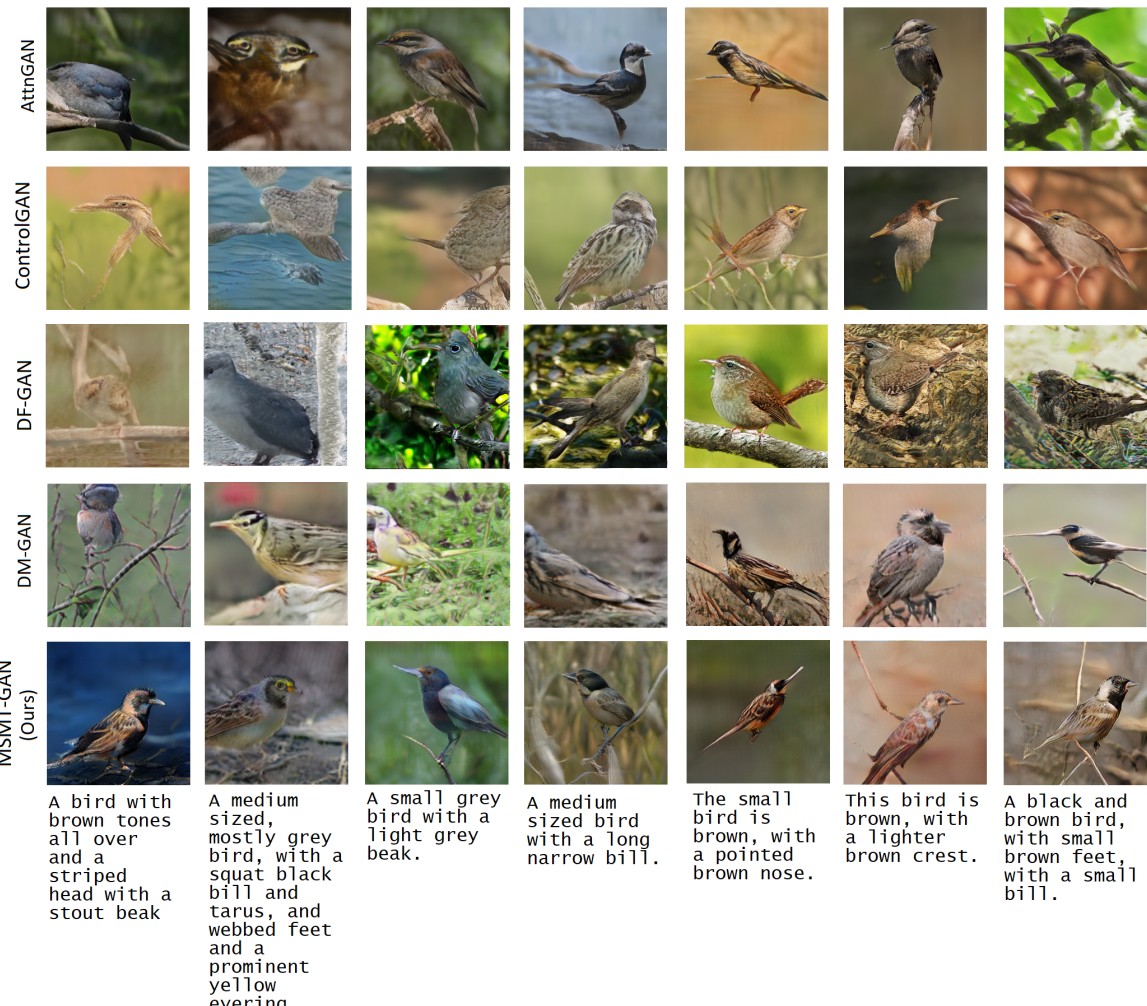

Figure 5: Extended comparison of MSMT-GAN with state-of-the art models on the CUB dataset.

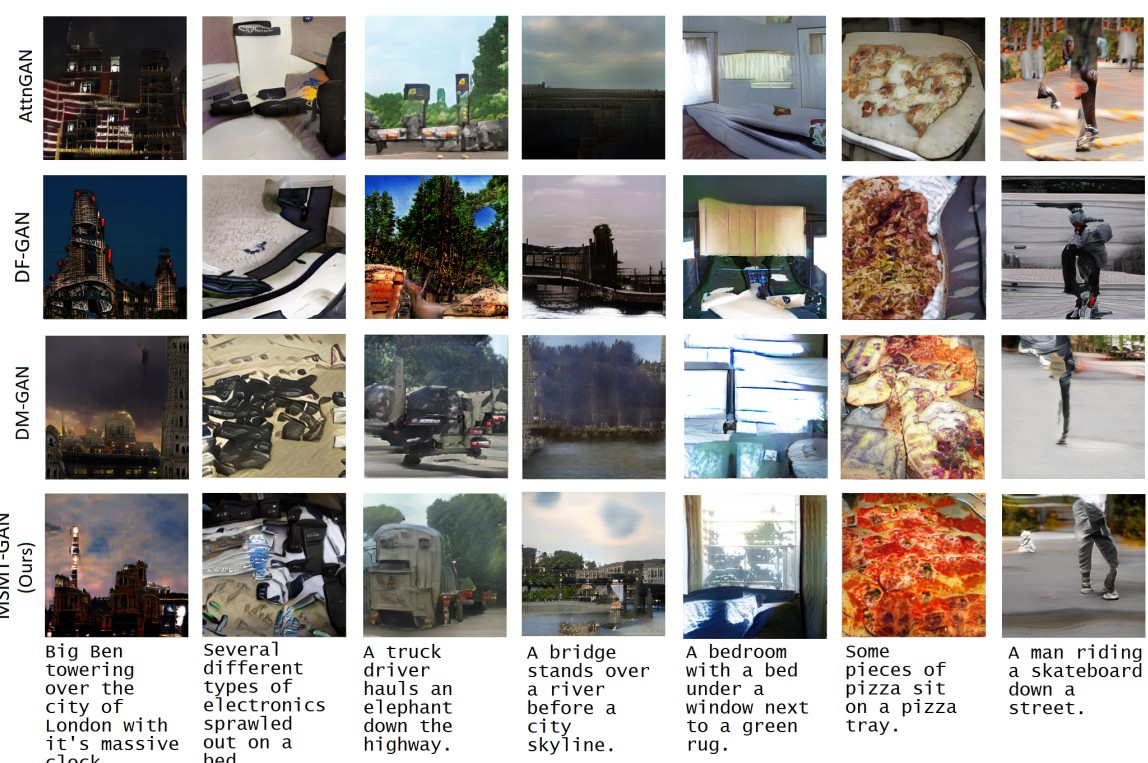

Figure 6: Extended comparison of MSMT-GAN with state-of-the art models on the COCO dataset.

