# OpenReview forum: "Multi-Tailed, Multi-Headed, Spatial Dynamic Memory Refined Text-to-Image Synthesis"
_TMLR — Withdrawn by Authors_

### Review · Reviewer_my8q · 2022-12-24

**Summary Of Contributions:**

The paper focuses on the text-to-image synthesis task with low-resource models. The authors first argue that there are three limitations of existing methods: (1) the initial stage only uses sentence-level features and does not use word-level features, (2) no constructing region-specific representations, and (3) limited precision in refinement because of using single-shot refinement. The paper proposes MSMT-GAN with three components (i.e., MTWIG, SDM, IMHM) to address each limitation, respectively. The experiments show that the proposed MSMT-GAN can improve performance against existing methods.

**Audience:**

No

**Broader Impact Concerns:**

The broader impact statement addresses the concerns well. There are no more concerns from my perspective.

**Claims And Evidence:**

No

**Requested Changes:**

Major requested changes that are important for acceptance:

* Clarify the claim on region-specific representations and the difference between SDM and other cross-attention methods.
* Clarify the effectiveness of MTWIG.
* Address the misleading descriptions of results.
* Evaluation metrics:
    * Replace R-Precision (Xu et al., 2018) with SOA (Hinz et al., 2020) on COCO and CLIP R-Precision (Park et al., 2021) on CUB.
    * Clarify if anti-aliasing is used for resizing when computing FID.
    * Add FLOPs, memory usage, and human evaluation results.
* Discuss or compare with related works that leverage unconditional StyleGAN for text-to-image synthesis.

Minor requested changes:

It is better to move the loss function from the Appendix (Section 7.4) to the main paper, especially with a new redundancy loss proposed by the authors.

Typos:

Implementation details in Section 4: Our objective Function -> our objective function


References

Hinz, Tobias, Stefan Heinrich, and Stefan Wermter. “Semantic Object Accuracy for Generative Text-to-Image Synthesis.” TPAMI 2020.

Park, Dong Huk, Samaneh Azadi, Xihui Liu, Trevor Darrell, and Anna Rohrbach. “Benchmark for Compositional Text-to-Image Synthesis.” in NeurIPS Datasets and Benchmarks Track (Round 1) 2021.


**Strengths And Weaknesses:**

Strengths:

* The paper is well-written and easy to follow.

* The code is attached for better reproducibility.


Weaknesses:


* **[The claim on region-specific representations and the difference between SDM and other cross-attention methods]** I am not convinced by the authors’ claim that existing methods were limited by not constructing region-specific representations. For example, in Figure 1 of AttnGAN, the authors show that the different words can attend to different spatial regions of the visual representations. ControlGAN proposes the word-level discriminator to encourage region-specific representations. DAE-GAN (Ruan et al., 2021) proposes Aspect-aware Dynamic Re-drawer for the same purpose. XMC-GAN also proposes “contrastive loss between image regions and words” based on the cosine similarity between words and image regions. On the other hand, while the authors claim the novelty of the proposed SDM module (see the summary of contribution in Section 1), it is still based on the cross-attention between text and visual representation like existing methods. Therefore, I am not convinced by the claim of the novelty of SDM.

* **[Ablation study of MTWIG]** In Table 1 and Section 4.1, the paper studied the “effectiveness of Multi-Tailed Word-level Initial Generation” by replacing the refinement method in MSMT-GAN with DM-GAN’s refinement method and comparing it against the vanilla DM-GAN method. However, an ablation study to remove or replace the component within the proposed method since different components can interact with each other. The current ablation study can show that the MTWIG can improve the results for DM-GAN instead of MSMT-GAN. Therefore, the effectiveness of MTWIG within the proposed MSMT-GAN needs to be well-demonstrated. Instead, the ablation study between IG and MTWIG in Table 4 (Section 7 Appendix) should be used as the ablation study for MTWIG. In Table 4, IG outperforms MTWIG in R-precision on CUB. At the same time, IG achieves very similar results (i.e., results have overlaps in standard deviation) in R-precision on the COCO dataset. I understand that MTWIG achieves better synthesis quality in FID. However, the motivation for proposing MTWIG is to generate more details with word-level information. Achieving better R-precision to achieve better image-text alignment should be the goal. In summary, the effectiveness of MTWIG is not well-demonstrated via experiments.

* **[Misleading Description of Results]** The claim that the proposed model is “comparable in size to previous methods” is misleading. In Table 2, the proposed MSMT-GAN has significantly more parameters than existing methods, e.g., ~2 times larger than DM-GAN. On the other hand, some results are bolded even with an overlap with the results of other methods (92.46+-0.28 in Table 3), which is misleading.

* **[Problems in Evaluation Metrics]** There are two problems regarding evaluation metrics. First, regarding the evaluation metrics used in the paper, R-Precision (Xu et al., 2018) should not be used anymore. (Zhang et al.) show that using image-text encoders to compute R-Precision skews the results: “many generated models report R-precision scores signiﬁcantly higher than real images.” The reason is that the image-text encoder was also used as the objective function—the DAMSM loss, also used by MSMT-GAN (Equation 19). Regarding FID, I appreciate the authors’ explanation of the result differences caused by PyTorch and Tensorflow implementation. However, (Parmar et al., 2022) show that anti-aliasing in image resizing plays a vital role in FID results, which is not specified in the paper. Second, some important evaluation metrics are missing in the paper. Regarding the “low-resource” aspect that the authors emphasized in the paper, I appreciate the authors reporting the number of parameters and number of GPUs in Table 2. However, the authors should also report the computational overhead in FLOPs and memory usage to better demonstrate the “low-resource” aspect of the model. Besides, there is no human evaluation to evaluate the photo-realism and image-text alignment, as in XMC-GAN (Zhang et al., 2021).

* **[Missing Related Work]** While the authors mainly studied the text-to-image models with low computational resources, the authors did not discuss or compare with the other line of works focusing on transforming the pretrained unconditional StyleGAN (Karras et al., 2019) into conditional text-to-image synthesis models, e.g., CI-GAN (Wang et al., 2021), TTF-HD (Wang et al., 2021), TediGAN (Xia et al., 2021), FuseDream (Liu et al., 2021), StyleMC (Kocasari et al., 2022), StyleT2I (Li et al., 2022), LAFITE (Zhou et al., 2022). These works can also be trained with a single GPU.


**References**

Ruan, Shulan, Yong Zhang, Kun Zhang, Yanbo Fan, Fan Tang, Qi Liu, and Enhong Chen. “DAE-GAN: Dynamic Aspect-Aware GAN for Text-to-Image Synthesis.” in ICCV 2021.

Xu, Tao, Pengchuan Zhang, Qiuyuan Huang, Han Zhang, Zhe Gan, Xiaolei Huang, and Xiaodong He. “AttnGAN: Fine-Grained Text to Image Generation With Attentional Generative Adversarial Networks.” in CVPR 2018.

Zhang, Han, Jing Yu Koh, Jason Baldridge, Honglak Lee, and Yinfei Yang. “Cross-Modal Contrastive Learning for Text-to-Image Generation.” in CVPR 2021.

Parmar, Gaurav, Richard Zhang, and Jun-Yan Zhu. “On Aliased Resizing and Surprising Subtleties in GAN Evaluation.” in CVPR 2022.

Karras, Tero, Samuli Laine, and Timo Aila. “A Style-Based Generator Architecture for Generative Adversarial Networks.” in CVPR 2019.

Wang, Hao, Guosheng Lin, Steven C. H. Hoi, and Chunyan Miao. “Cycle-Consistent Inverse GAN for Text-to-Image Synthesis.” In ACM Multimedia Conference on Multimedia Conference - MM 2021.

Wang, Tianren, Teng Zhang, and Brian Lovell. “Faces a La Carte: Text-to-Face Generation via Attribute Disentanglement.” in WACV 2021.

Xia, Weihao, Yujiu Yang, Jing-Hao Xue, and Baoyuan Wu. “TediGAN: Text-Guided Diverse Face Image Generation and Manipulation.” in CVPR 2021.

Liu, Xingchao, Chengyue Gong, Lemeng Wu, Shujian Zhang, Hao Su, and Qiang Liu. “FuseDream: Training-Free Text-to-Image Generation with Improved CLIP+GAN Space Optimization.” ArXiv:2112.01573 [Cs], 2021.

Kocasari, Umut, Alara Dirik, Mert Tiftikci, and Pinar Yanardag. 2022. “StyleMC: Multi-Channel Based Fast Text-Guided Image Generation and Manipulation.” In The IEEE Winter Conference on Applications of Computer Vision (WACV).

Li, Zhiheng, Martin Renqiang Min, Kai Li, and Chenliang Xu. “StyleT2I: Toward Compositional and High-Fidelity Text-to-Image Synthesis.” in CVPR 2022.

Zhou, Yufan, Ruiyi Zhang, Changyou Chen, Chunyuan Li, Chris Tensmeyer, Tong Yu, Jiuxiang Gu, Jinhui Xu, and Tong Sun. “LAFITE: Towards Language-Free Training for Text-to-Image Generation.” in CVPR 2022.

---

### Review · Reviewer_jgDJ · 2023-01-04

**Summary Of Contributions:**

This paper tackles text-to-image generation with limited GPU resource.

The proposed method introduces 1) a word-level initial stage, 2) spatially varying words, and 3) an iterative multi-head mechanism for image refinement.


**Audience:**

Yes

**Broader Impact Concerns:**

Fine.

**Claims And Evidence:**

No

**Requested Changes:**

Please check the weakness part.

**Strengths And Weaknesses:**

Strengths
1. Ablation study demonstrates effectiveness of spatial dynamic memory and the iterative multi-head mechanism.
1. The proposed method outperforms existing methods with <100M params.

Weaknesses
1. Why do we need a low-resource model with inferior performance? An alternative choice would be starting from high-resource models with superior performance and then fine-tuning or introducing some sort of domain adaptation. It is a common approach in the literature [stylegan2-ada, eg3d, diffusionclip]
1. Is stable diffusion larger than 100M params?
1. Why is the memory mechanism suitable for the refinement head? 3.1 and 3.3 start with briefly explaining the limitations of existing methods but 3.2 does not.
1. Why is iterative refinement always better than single-shot refinement?
1. Figure 1 does not match the corresponding description and has unnecessary complexity. E.g., fuse component can be simply one box without the inner circuit.
1. Would the proposed method improve when combined with T5 or CLIP text encoder?
1. The datasets are outdated: CUB and COCO. Please consider using LAION.
1. Improvements in Table 1 are minor.
1. Values in the last row of Table 1 and CUB in Table 3 are different. Mistake?
1. How do we know whether MTWIG really reflects the words or not?
1. I wonder how much the multiple refinement steps actually helps. Would the image prediction produce gradually improving images over the refinement steps?
1. Writing should be clearer and more compact. Please separate long sentences and trim out unimportant descriptions.

< Minor >
However, current methods suffer three important limitations.
-> However, current methods suffer from three important limitations.

… words-level details are are inherently …
-> … word-level details are inherently …

… and point out that that accounting for …
-> … and point out that accounting for …

---

### Review · Reviewer_DbX9 · 2023-02-27

**Summary Of Contributions:**

The papers main contribution is a relatively small text-to-image model (<100M parameters) which produces qualitatively and quantitatively better images on the Caltech-UCSD Birds 200 (CUB) dataset and minor improvements on MS-COCO.


**Audience:**

Yes

**Claims And Evidence:**

Yes

**Requested Changes:**

Overall the paper is well organized, well written and straightforward to follow. The claims presented are backed by convincing experiments, e.g. showing that the individual components the authors add each actually contribute to improving image synthesis performance on CUB.

My two main criticisms and requests for change are the following:

1. Definition of "small": The 100M parameter boundary seems reasonable, but at the same time just too convenient for the authors' analysis. While their own model is twice as big as the next best in their list of competitors (DM-GAN), but the 111M parameter XMC-GAN, which itself is roughly twice as big as the authors' model is excluded because it is >100M parameters. It's quite clear that there is a strong correlation between model size and performance, so the key question is whether the authors' model is particularly efficient or just on the same line as most previous models. I think rather than defining arbitrary cutoffs, it would make more sense to present the results as a plot with number of parameters on the x axis and performance on the y axis and include also bigger models such as XMC-GAN. Such a presentation of the data would give the reader a better impression of the model landscape. Given that XMC-GAN is quite substantially better on COCO (FID ca. 10 vs. 23 for the authors' model), I suspect that the authors' model actually does not really present an improvement over the state of the art on COCO.

1. Improvements mostly limited to CUB: Given the point above, I think the model mostly presents an improvement on the CUB dataset. I think this is totally fine and the authors also acknowledge this fact to some extent in the paper. However, there is no mention of the scope of the contributions in the abstract, which sounds much more general than the paper actually delivers. I suggest to fix that and mention already in the abstract the focus on CUB.

**Strengths And Weaknesses:**

### Strengths

+ Paper is straightforward to read and presents evidence for the claims it makes
+ Improving on "small" models that can be trained on a single GPU is a worthwhile goal


### Weaknesses

- The definition of "small" (<100M parameters) seems a bit arbitrary and "convenient" for the authors' claim
- Improvements mostly limited to the birds dataset; generality less clear

---

### Note · Authors · 2023-03-13

**Comment:**

We would like to thank the Reviewers my8q, DbX9, jgDJ and action-editor very much for taking the time to read and review our paper. However, we find that we are not able to conduct the requested additional experiments and paper-revisions within the time-frame the TMLR deadline, and have therefore decided to withdraw our submisison from consideration. Thank you again for for your comments - we will aim to address this feedback in subsequent paper revisions.

**Withdrawal Confirmation:**

I have read and agree with the venue's withdrawal policy on behalf of myself and my co-authors.